# Cognitive and Behavior Deficits in Parkinson’s Disease with Alteration of FDG-PET Irrespective of Age

**DOI:** 10.3390/geriatrics6040110

**Published:** 2021-11-10

**Authors:** Fulvio Lauretani, Livia Ruffini, Crescenzo Testa, Marco Salvi, Mara Scarlattei, Giorgio Baldari, Irene Zucchini, Beatrice Lorenzi, Chiara Cattabiani, Marcello Maggio

**Affiliations:** 1Department of Medicine and Surgery, University of Parma, 43124 Parma, Italy; crescenzo.testa@unipr.it (C.T.); marco.salvi@unipr.it (M.S.); irene.zucchini@unipr.it (I.Z.); beatrice.lorenzi@unipr.it (B.L.); chiarac2004@libero.it (C.C.); marcellogiuseppe.maggio@unipr.it (M.M.); 2Geriatric Clinic, Geriatric-Rehabilitation Department, University Hospital, 43124 Parma, Italy; mscarlattei@ao.pr.it (M.S.); gbaldari@ao.pr.it (G.B.); 3Nuclear Medicine Division, University-Hospital of Parma, 43124 Parma, Italy; lruffini@ao.pr.it

**Keywords:** molecular imaging, ICDs, 18F-FDG-PET, Parkinson’s disease, dopamine agonists

## Abstract

Significant progress has been made in our understanding of the neurobiology of Parkinson’s disease (PD). Post-mortem studies are an important step and could help to comprehend not only the progression of motor symptoms, but also the involvement of other clinical domains, including cognition, behavior and impulse control disorders (ICDs). The correlation of neuropathological extension of the disease with the clinical stages remains challenging. Molecular imaging, including positron emission tomography (PET) and single photon computed tomography (SPECT), could allow for bridging the gap by providing in vivo evidence of disease extension. In the last decade, we have observed a plethora of reports describing improvements in the sensitivity of neuroimaging techniques. These data contribute to increasing the accuracy of PD diagnosis, differentiating PD from other causes of parkinsonism and also obtaining a surrogate marker of disease progression. FDG-PET has been used to measure cerebral metabolic rates of glucose, a proxy for neuronal activity, in PD. Many studies have shown that this technique could be used in early PD, where reduced metabolic activity correlates with disease progression and could predict histopathological diagnosis. The aim of this work is to report two particular cases of PD in which the assessment of brain metabolic activity (from FDG-PET) has been combined with clinical aspects of non-motor symptoms. Integration of information on neuropsychological and metabolic imaging allows us to improve the treatment of PD patients irrespective of age.

## 1. Introduction

Parkinson’s disease (PD) is an age-related neurodegenerative disorder that affects as many as 1–2% of people aged 60 years and older [1]. The incidence and prevalence of this disease are constantly growing, mainly due to the aging of the population [2]. It has been estimated that the annual incidence ranges from 5/100.000 to over 35/100.000 new cases, with a 5–10-fold increase from the sixth to the ninth decade of life [3,4]. The prevalence of the disease also increases with age. In a meta-analysis of four North American populations, the prevalence of the disease was shown to increase from nearly 1% in the first 55 years of life to over 4% in men over 85 years of age.

Lower prevalence values were found in women with “only” 2% of women affected by the disease when over 85 years of age [5]. Despite the serious impact on the quality of life of patients, which can be mitigated by the treatments that will be discussed later, the mortality in these patients is equal to those unaffected by PD in the first ten years from diagnosis, and tends to increase later [6].

It is evident that in older populations, which are more predisposed to developing neurodegenerative diseases such as Parkinson’s disease, the social and economic aspects deriving from this epidemiological framework will be increasingly important [7,8]. Therefore, in the therapeutic approach of this disease, it is fundamental to avoid treatments that can contribute themselves to lowering the quality of life, not only of patients, but also of caregivers who are central figures in the standards of care of this disease [9].

Impulse control disorders are one of the side effects that most negatively impact the quality of life of patients and their caregivers under dopamine replacement therapies. Other disorders including pathological gambling, compulsive sexual behavior and buying, and binge or compulsive eating have been described [10]. These disorders generally affect younger patients and those who have first-degree relatives in the family in whom similar problems have been described, regardless of the diagnosis of Parkinson’s disease [11]. Although ICDs have been associated with high dose levodopa treatment, it is important to note that this side effect is significantly more pronounced in patients receiving dopamine agonists [11]. The reason for this association is unclear, but one of the most probable hypotheses is based on the affinity profile of dopaminergic agonists for some dopamine receptor isoforms. In fact, while the motor effects of dopamine replacement therapies are due to dopamine D1 and D2 receptors, ICDs are due to an effect on dopamine D3 receptors [12]. D3 receptors are mostly expressed in the ventral striatum [12], a prosencephalic region associated with both behavioral addictions [13] and substance abuse disorders [14].

It has been shown that especially new dopaminergic agonists (ropinirole and pramipexole) have greater selectivity for D3 receptors than for D1 and D2 dopamine receptors [15]. It is therefore evident that we are no longer faced, as has been hypothesized for a long time, with a disease with only implications on motor functions. In 2003, Braak and colleagues already hypothesized a neuroanatomic staging of the disease affecting different domains (Braak’s stages 1 and 2 are associated with the premotor phase; Braak’s stages 3–4 are associated with the motor symptoms; and Braak’s stages 5–6 are associated with cognitive impairment) [16].

Taking into account the complexity of symptomatology of PD, it becomes important to plan an appropriate treatment to balance motor and cognitive and behavior symptoms of the patient [17]. 

In fact, L-dopa or dopamine agonists may produce neuropsychiatric side effects not only in relation of the age of the patient but also if the patient presents some degree of cognitive impairment or psychiatric symptoms such as ICDs.

PET with 18F-fluorodeoxyglucose (FDG-PET) has been used to measure regional cerebral glucose metabolism, a proxy for neuronal activity, in PD. Many studies have shown that this technique could be used in early PD, where reduced metabolic activity correlates with disease progression and predict histopathological diagnosis [18].

The clinical usefulness of introducing PET-FDG in the therapeutic work-up of Parkinson’s disease is not only related to the increase in sensitivity and diagnostic specificity, but also in improving the stratification of patients, thus increasingly promoting personalized medicine.

First of all, the PET-FDG describes hypometabolism patterns typical of Parkinson’s disease (Parkinson’s Disease Related Pattern, PDRP) [17]. Furthermore, unlike other techniques, PET-FDG offers high sensitivity with admirable spatial and temporal resolution. It is also important to underline that unlike the DAT-scan, PET-FDG is able to highlight hypometabolisms not only in the dopaminergic pathways, but also in other receptor pathways upstream or downstream of these. In this way, while keeping in mind the fundamental importance of dopaminergic pathways in the pathogenesis of Parkinson’s disease, it is also possible to investigate other receptor pathways certainly involved in the development of non-motor disorders [19].

The aim of this work is to report on particular cases of PD in which assessment of cerebral metabolic activity (from FDG-PET) has been combined with clinical evaluation of non-motor symptoms. Integration of clinical and metabolic imaging information provide to better treat patients with PD irrespective of age.

## 2. Methods

Patients were evaluated at the Cognitive and Motor Ambulatory of the Geriatric-Rehabilitation Department of the University Hospital of Parma, while the FDG-PET scans were performed at the Nuclear Medicine Department of University Hospital of Parma (Parma, Italy). Patients were first evaluated by an expert geriatrician with a standard clinical evaluation [20,21] and then referred to a psychologist with long-term experience in the evaluation of older persons because of complaints of neuro-psychiatric symptoms. Diagnosis of PD was made according to Movement Disorders Society (MDS) criteria [22], while all cognitive tests included in the neuropsychology battery have norms and cut-offs available for the Italian population [21]. The 15 item Geriatric Depression Scale (GDS) was used for assessing depression [23] and the Wisconsin Card Sorting Test was used to examine aspects of executive function [21]. All patients had a recent MRI or CT scan. All drug consumption was recorded. The two patients gave the consent to record these data. They are enrolled in the prospective observational T.R.I.P. Study (Traumatic Risk Identikit Parma Study). (The study was approved by the Ethical Committee of the University Hospital of Parma, ID 17262 del 12/05/2017). Data were treated in agreement with Italian law for the privacy guaranty.

### 2.1. 18F-FDG PET/CT

An 18F-FDG scan was performed using a whole-body hybrid system Discovery IQ (GE Healthcare) operating in three-dimensional detection mode. A head holder was used to restrict patient movement and head movement was checked on a regular basis. 

After an overnight fast, 200 MBq of 18F-FDG was administered intravenously in a quiet, dimly lit examination room. The brain PET/CT recording was started 30 min after tracer injection. During the 30-min uptake period, participants were left undisturbed in a darkened room and instructed to rest quietly without activity with their eyes closed, as is commonly recommended. The brain CT was first recorded to provide the attenuation correction map (140 kV, 25 mA, 512 × 512 matrix, 3.75-mm slice thickness, scan Type Helical full 0.8 s, No of images 79, Rec Fov 30 cm, recon type standard). CT was immediately followed by a 3D-PET recording during a 10-min period (FOV 30 cm, recon type QCHD and VPHD, matrix size 256 × 256). Quantitative analysis was performed using the SPM5 software implemented in Matlab R2014a [24]. 

The patient PET dataset is spatially normalized using the SPM5 PET template and smoothed with a Gaussian filter of 8 mm FWHM. Differences in CMRglc (patient vs. normal) are assessed on a voxel-by-voxel basis, using a paired t-test. The results are displayed on the Tailarach atlas.

### 2.2. Example of Young and Older PD Patient Showing ICDs after Using Dopamine Agonists: Evaluation with FDG-PET

#### 2.2.1. First Case (Young Patient: PD with ICDs after Using Ropinirole)

A 55-year-old man with resting tremor and rigidity was evaluated at the Cognitive and Motor Ambulatory, University Hospital of Parma. His medical history was characterized by presence of diabetes, hypertension and chronic ischemic coronaropathy. At the first visit a resting tremor at the right hand and plastic rigidity with “lead-pipe” rigidity was observed. Given the multimorbidity of the patients, a SPECT with 123I-ioflupane (DaTSCAN^©^) (Figure 1) was prescribed. The exam showed reduced uptake in dopamine making the diagnosis of PD probable, according to the MDS criteria. Mini Mental Examination State (MMSE) score was normal (30/30 corrected for age and education) as well as cerebral MRI. Initially, a standard dose of dopamine agonists, ropinirole extended release was prescribed, and even if motor symptoms one year later were reduced, the patient’s wife reported a strange behavior, including excess of shopping and changes of sexuality with clear hypersexuality.

Given psychiatric symptoms reported, we changed the current treatment starting L-dopa and prescribed the FDG-PET, because of the severe anxiety induced in family members of this young PD patient. FDG-PET showed extended hypometabolism in the right and left inferior frontal gyrus and in part of the prefrontal lateral areas (Figure 2). A further cognitive assessment by MMSE remained normal and even the second level of neuropsychological evaluation resulted normal. After the prescription of L-dopa at the dosage of 300 mg daily, motor symptoms disappeared and patient’s wife reported a profound improvement of behavior, with disappearance of ICDs. 

#### 2.2.2. Second Case (Older Patient: PD with ICDs after Using Pramipexole)

A 70-year-old man with established diagnosis of PD suggested by a private neurologist, was evaluated at the Cognitive and Motor Ambulatory, University Hospital of Parma, two years after the diagnosis. All cardinal motor symptoms were present, with an evident association of depression as non-motor symptom. The patient used pramipexole as first drug prescribed by neurologist, but he reported a motoric difficult to perform activities of daily living. Thus, he asked us to revise dosage or change the drug. He was retired from his principal work, but he was still involved as a lecturer in theology. At the initial cognitive assessment, he showed a normal MMSE (30/30 corrected for age and education). The evaluation included a structured neurological visit without evidence of neurological clinical signs. The patient’s wife reported an increase of sexuality, completely unexpected in relation to the past behavior and given his catholic principles. The patient received the second-stage cognitive screening by an expert psychologist and logopedist and the evaluation resulted normal.

Given the reported ICDs, we changed treatment using L-dopa and prescribed the FDG-PET, to exclude other forms of parkinsonism. The FDG-PET scan showed hypometabolism in many cerebral areas involving, also in this case, the right and left inferior frontal gyrus and part of the prefrontal lateral areas (Figure 3). After the prescription of L-dopa at the dosage of 400 mg daily, we also added venlafaxine for improving depressive symptoms. After a 5-year follow-up period, the patient reported a discrete control of motor symptoms, e.g., he drove his car for more than two hours, and his wife reported a completely disappearance of ICDs, with significant improvement of behavior returning to the same as before starting treatment.

## 3. Discussion on the Integration of Neuropsychological and Neuroimaging Information

The dopaminergic system is mainly composed in two ways: 1. the Nigro-Striatal system, rich in D1 and D2 dopaminergic receptors, which are responsible of the movement; and 2. Meso-Limbic and prefrontal cortex-ventral striatal systems, where are mainly found D3 and D4 dopaminergic receptors, which are responsible for the positive symptoms of schizophrenia such as hallucinations and delusions. Direct consequences of alteration of these pathways, and in particular of the “prefrontal cortex-ventral striatal systems”, when using dopamine agonists, could be the developing of ICDs even in an initial phase of the disease. Previously, we have shown that dopamine agonists have a greater selectivity for D3 receptors [15]. A recent study showed that treatment with dopamine agonists interferes with the functional activity of the prefrontal cortex and the ventral striatum [25]. Regardless of dopamine agonist treatment, in patients who had a greater tendency to develop ICDs there was less functional activity in the right inferior frontal gyrus [25]. The importance of the inferior frontal gyrus in the inhibition of impulsive controls was already described in the literature. The presence of lesions in this region, regardless of Parkinson’s disease, can cause ICDs [26].

All these data support what was reported by Ruppert MC and colleagues, who showed that high impulsivity, defined as a personality trait, predisposes to risk-seeking behavior, and is closely associated with impaired FDG-metabolism within the fronto-insula network in PD [27].

ICDs seem to be caused by a heterogeneous melting pot of predisposing factors independent of Parkinson’s disease together with an altered response to dopamine agonist therapies.

The development of an ICD can be compared with an “epileptic seizure” where different triggers (dopamine agonist treatment) work above an epileptogenic substrate (personal trait of developing ICD, in particular a reduced activity in the inferior frontal gyrus). In this direction, the utilization of FDG-PET to detect a metabolic alteration in the inferior frontal gyrus could orient clinician’s in selecting the most appropriate drug for PD treatment.

Over the years, an increasing number of physicians started to use dopamine agonists in younger patients confining levodopa treatment to the most advanced stages of the disease [28].

However, early levodopa treatment, in addition to be effective at all patient’s age, does not interfere with the progression of the disease [29,30]. Although dopamine agonists predispose to side effects such as ICDs [31], their use remains fundamental given the well-known and greater risk of developing dyskinesias induced by levodopa.

Correct stratification of patients in the initial stages of the disease is therefore essential to decide the most appropriate treatment. The presented case reports permit to support the use of molecular imaging with FDG-PET for selecting drugs in the early phase of PD, even in patients with a medical history negative for significant psychiatric symptoms.

The limitations of our study may be to suggest a PET scan with relatively high costs, when the full clinical evaluation could be realized with an exhaustive explanation of the side-effects of dopamine agonists. However, FDG-PET could reinforce reasons why these medications produce ICDs only in some patients. Furthermore, our report is based on data from only two patients. It would be improper to generalize our conclusions to the general population: for this reason, further studies are needed to investigate whether our conclusions can be generalized to the general population. It would be important, in the diagnostic/therapeutic flow-chart of Parkinson’s disease, a method that not only increases the sensitivity and specificity in making the diagnosis, but that stratifies patients in order to propose a treatment as personalized as possible, so as to propose an increasingly personalized medicine. From this perspective, the cost effectiveness of proposing a diagnostic investigation that still remains expensive would also improve.

Many therapeutic options have become available for PD in recent years, leading to significant improvements in motor control both at early and advanced disease stages [31]. The need to expand disease management beyond motor symptoms control has been recently highlighted [32]. There is recent evidence that on-motor features deeply affect the quality of life of patients [33].

Dopamine agonists represent a valid therapeutic option in PD and their effects on non-motor domains like mood or cognition are acknowledged as key factors in fully addressing patients’ needs.

The balance between motor deficits and cognitive or psychiatric symptoms seems to be the most important factor contributing to treatment decision-making when approaching PD therapy.

In conclusion, earlier [17] and more accurate diagnosis of PD with impaired all dopaminergic pathways may help to improve patient’s health status and reduce treatment costs by effectively allocating healthcare resources and maximizing the benefit of treatments and supportive services.

Use of FDG-PET may help to accurately identify PD patients that could develop ICDs after using dopamine agonists.

Unlike previous works [19], our approach is not so much aimed at demonstrating the diagnostic utility of PET-FDG in Parkinson’s disease, but is aimed, instead, at a better stratification of patients so as to prevent serious side effects that, in predisposed patients, can lead to serious consequences. Furthermore, with PET-FDG it is possible to discriminate certain hypometabolism patterns significant for cognitive deterioration. Recent evidence indicates that these patients are also more predisposed to the development of ICDs [34].

The development of novel model of clusters of PD patients [35] may provide additional benefits by slowing or halting progressive decline of PD, increasing quality of life and prolonging survival.

## Figures and Tables

**Figure 1 geriatrics-06-00110-f001:**
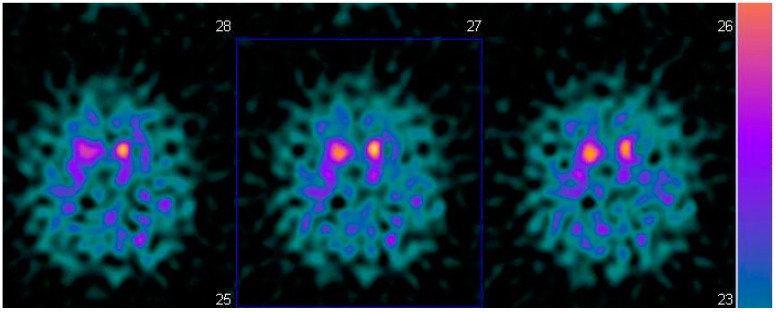
Brain SPECT with 123 I-iofuplain (DaTSCAN).

**Figure 2 geriatrics-06-00110-f002:**
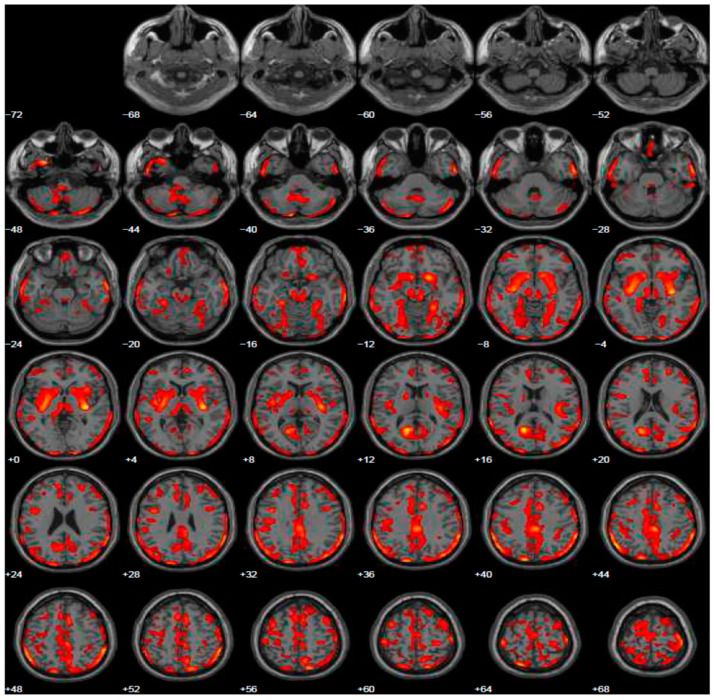
Brain PET with 18F-FDG. Images highlight brain regions consistently found the analysis (Statistical Parametric Mapping software SPM 5 (*p* = 0.05). Diffuse hypometabolism in the left middle temporal gyrus, right and left inferior frontal gyrus, left middle frontal gyrus, left posterior cingulate, left inferior parietal lobule, right lentiform nucleus (putamen), right caudate head, left thalamus, left parahippocampal gyrus, right red nucleus (midbrain), left cerebellum pyramis, right and left cerebellum inferior semi-lunar lobule, right and left cerebellum posterior lobe tuber, right cerebellum anterior lobe nodule, left cerebellar tonsil.

**Figure 3 geriatrics-06-00110-f003:**
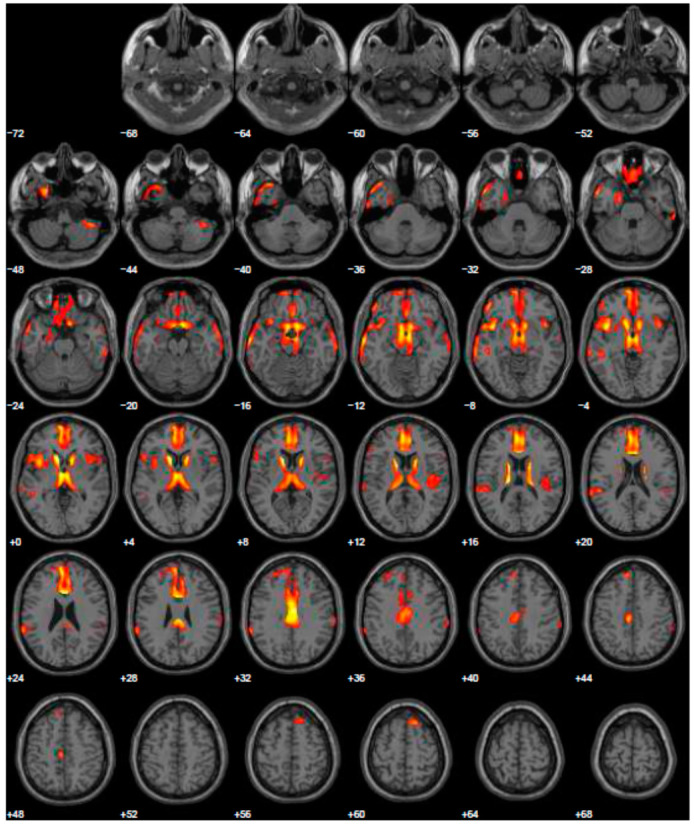
Brain PET with 18F-FDG. Images highlight brain regions consistently found the analysis (Statistical Parametric Mapping software SPM 5 (*p* = 0.05). Hypometabolism in the left caudate body and head, left middle temporal gyrus, left superior temporal gyrus, right superior frontal gyrus, left and right inferior parietal lobule, left and right cerebrum sub-lobar insula, left and right superior temporal gyrus, left and right inferior frontal gyrus, right sub-lobar extra-nuclear gray matter, right cerebellar tonsil, right inferior temporal gyrus, left and right middle temporal gyrus, left limbic lobe uncus, left limbic lobe parahippocampal gyrus, right transverse temporal gyrus.

## Data Availability

Not applicable.

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
