# Peer review of "Cognitive and Behavior Deficits in Parkinson’s Disease with Alteration of FDG-PET Irrespective of Age"

_geriatrics, 2021, doi:10.3390/geriatrics6040110_

Round 1
Reviewer 1 Report
Thanks for recommending me as a reviewer. This paper were aim to report two particular cases of PD in which the assessment of brain metabolic activity (from FDG-PET) has been combined with clinical aspects of non-motor symptoms. If the authors complete the minor revision, the quality of the study will be further improved.
- The introduction section is well written. If the authors describe in more detail the trends on the effectiveness of FDG-PET treatment for behavioral interventions in Parkinson's disease in the introduction section, it may help readers to understand.
- line 102-119: Authors should check the font size in this paragraph.
- Authors should add study limitations to the discussion section.
Author Response
Dear Reviewer
Attached reply to your comments.
Best regards
Fulvio Lauretani

Reviewer 2 Report
This case report described the effectiveness of FDG PET in the treatment of PD patients. I read this report with great interest. I felt the novelty was ambiguous in current form, and I think there is a gap in logic, if you generalize the functional imaging results obtained from only two patients.
- I compare this report and previous review (Z Walker, et al. Eur J Nucl Med Mol Imaging. 2018 Jul; 45(9): 1534-1545.), and I think you should clarify the novelty of your report.
- DaTscan and MIBG scintigraphy reflect the pathology and clinical condition better than FDG-PET. If you discus about the usefulness of FDG-PET, please compare the effectiveness of FDG-PET with DaTscan and scintigraphy.
- Please make sure the citation is appropriate. I think there are many unnecessary descriptions in the method part.
For example
・Ref 19 (line 105) and Ref 20 (line 107), necessary?
・Is the GDS you wrote in the method part (line 111) mentioned afterwards?
・I think Ref 22 is more suitable for J A Yesavage, et al. J Psychiatr Res. 1982-1983;17(1): 37-49.
Author Response
Dear Reviewer
Attached reply to your comments.
Best Regards
Fulvio Lauretani

Round 2
Reviewer 2 Report
I think the manuscript has improved.